# Outdoor sports and active tourism company management in Cordoba (southern Spain): An empirical study on the perception and behavior of supply

Francisco Molina Navarro[1], Manuel Rivera Mateos[2], María Genoveva Millán Vázquez de la Torre[3]*

1 Department of Agrarian Economy, Cordoba University, Cordoba, Spain, 2 Department of Geography, Cordoba University, Cordoba, Spain, 3 Department of Quantitative Methods, Universidad Loyola Andalucía, Cordoba, Spain

* gmillan@uloyola.es

## Abstract

This paper analyzes the perception and behavior of active tourism companies that provide outdoor physical sports activities in emerging tourist destinations in Cordoba, an inland province of southern Spain. The active tourism supply of registered businesses was analyzed using fieldwork data and more than 30 variables including training and qualification of human resources, business vision of demand, marketing and promotion strategies, and perceptions of supply trends. To this end, a quantitative-qualitative empirical study was performed together with in-depth interviews of key stakeholders. Official registry sources were also used to complement the data. The results show a fragmented network of micro-SMEs that are functionally isolated from the rest of the tourism supply and display an intuitive and poor market-oriented behavior due to the inadequate management of their client portfolio. Such management models make it difficult for these businesses to take advantage of key strategic assets to increase profitability and achieve a competitive and sustainable advantage.

## Introduction

Contact with nature and the practice of outdoor physical sports activities are an emerging trend in tourism supply to meet postmodern society's growing demand for active recreation, leisure, and sports (traditional sports that have been adapted and new, alternative ones). These activities often involve a hedonistic, emotional, experiential, and/or adventurous component that may entail a certain amount of risk and surprise [1]. In Spain, outdoor sports tourism (hereinafter OST), or active tourism (the term *active tourism* (*turismo activo*) is used in its broadest sense in Spain in both the commercial tourism and the policy/administration sphere. The two terms will be used interchangeably in this paper), represents an excellent opportunity for economic development, employment, and production diversification in rural, mountainous, and inland environments [2] by bridging the socioeconomic gap between rural and urban

**Funding:** The authors have not received specific funding for this work.

**Competing interests:** The authors have declared that no competing interests exist.

populations. Moreover, this tourism subtype also serves as an important means for retraining, repositioning, and diversifying the mature product of many coastal sun, sand, and sea destinations [3, 4].

However, comprehensive and detailed analyses of this tourism product are still needed in many Spanish tourist destinations, not only to increase knowledge and make strategic business decisions in this subsector, but also to guide planning and management policies, which have been initiated only tentatively in most regions of Spain. Similarly, such analyses are important to ensure the sustainable management of regional and heritage resources, particularly the physical environment and landscapes that host and provide the proper conditions for such activities to take place. Studies on OST should also serve as a reference and increase knowledge of the environment with a view to implementing business intelligence measures that improve the competitiveness of this subsector and promote the development of new business projects.

Despite the numerous research advances, contributions, and case studies carried out in Spain in recent decades, a solid theoretical framework that delimits and defines sports and recreational tourism is still lacking. Additionally, a theoretical and conceptual framework is urgently needed as a cognitive basis for the empirical application of business studies in this subsector of emerging tourism activity and to develop theoretical models of supply behavior in specific regions or territories.

The scarce literature on OST supply in Spain is a major drawback for extrapolating findings to different areas and developing models for the dissemination, development, and consolidation of this business subsector. Rather than a stand-alone tourism product, this subsector should be considered an integral component of the overall tourism supply available in many destinations [5], and even as an important primary or secondary pull factor for demand. Moreover, the lack of empirical case studies makes it very difficult to establish a conceptual and terminological framework that could serve as an analytical tool and applied to diverse contexts and categories of study where a general overview of the situation is required. This scarcity of empirical literature is partly due to the relatively recent commercial and business supply of active tourism products in Spain. However, it is still somewhat surprising considering that the popularity of active tourism or OST has grown extraordinarily since the late 1980s [6] parallel to the development of the tourism industry and the increasing awareness of nature conservation and enjoyment as well as the health benefits of physical exercise [7]. In fact, the OST business supply in Spain has seen an almost exponential growth over the past two decades, as reflected in the large number of active companies in the country [8, 9].

Active tourism was given little attention until a couple of decades ago when it began to attract greater interest as more than a recreational or complementary component of other tourism activities and became a specific and differentiated tourist typology capable of creating niche destinations and unique supply initiatives for tourism spaces [2]. Interest in OST also grew due to its associated economic impact, the appeal of its values, and what it represents (contact with nature, ecology, well-being, enjoyment of the landscape, action and adventure, emotions associated with the use of certain types of equipment, freedom, etc.), and because sports can serve to build a brand image and promote many tourist destinations.

It is important to note that the research on active tourism has been carried out from very diverse approaches and angles, among them economic and business management, geography, marketing, sports sciences and motor behavior, sports psychology, quality management, sociology and anthropology, and an institutional perspective applied to the planning and development of this business subsector [10–13].

Nonetheless, when examining the OST phenomenon, a fundamental question arises that is neither easy to answer nor for which there is consensus: What are we referring to when we talk about this tourism product typology? The concept of active tourism, as it is referred to in the

Spanish tourism jargon instead of OST, encompasses a diverse tourism, sports, and recreational activities practiced in a myriad of scenarios with countless techniques and materials and a wide range of expert personnel. OST practices and travel include those tailored to families and groups of friends, company incentives, environmental education centers, youth camps, and federated or "adventure" sportsmen and women who are looking for an alternative to occupy their free time or spend their vacations. It is, therefore, a generic concept that covers an endless number of outdoor activities and sports tourism modalities. Very often, active tourism is a complementary product linked to several thematic tourism categories (rural; nature; cultural; sun, sea, and sand; winter and mountain; business meetings, etc.) and different types of accommodation. But it also constitutes a specialized product that is the primary motivation for travel (sports tourism of a specific type), particularly to niche destinations with a consolidated brand image (rafting in the Sierra de Guara of Aragon, surfing and kitesurfing in Tarifa, Cadiz, kayaking and canyoning in the eastern part of Asturias, winter sports in mountain resorts in Aragon, Catalonia, and the Sierra Nevada of Granada, among others). It should also be noted that active outdoor leisure involves short but frequent trips (i.e., hiking) from the city to peri-urban spaces of a certain environmental and recreational value with recently developed facilities such as adventure parks, zip lines, and hiking circuits, or to nearby rural and protected areas, which hold huge potential for the future demand of OST products.

As a category of active sports tourism [14, 15], OST consists of a set of active physical sports practices and "controlled adventure," which, as a commercialized product with a strong recreational focus, are practiced outdoors during free time in contact with nature, often using the resources provided by nature, and assuming a certain risk factor, which requires physical effort and/or skill by those who practice them. Active tourism is also very heterogeneous and comprises an almost unlimited range of modalities due to the diversity of resources available in the natural environments where the activities take place. The need for different types of sports equipment, the level of risk, skill, user capacity or physical preparation, as well as the techniques and types of activities have given rise to different sports cultures or subcultures [16]. Moreover, traditional outdoor sports have been segmented into multiple branches and adaptations, as is the case of skiing and mountaineering, which has resulted in more than twenty disciplines that can be classified into three groups according to where they are practiced: land sports (hiking, horseback riding, alpinism and mountaineering, mountain biking, spelunking, etc.); air sports (hang gliding, paragliding, hot-air ballooning, etc.), and water sports (whitewater rafting, canoeing, kayaking, diving, surfing, sailing, etc.).

As mentioned, *active tourism* is the conceptual term used in Spain and other Spanish-speaking countries to refer to these activities [9]. However, this might not be the most appropriate term given that it is similar or almost synonymous with *activity holidays*; a broad, generic term that refers to a wide range of activities including outdoor and non-motorized sports, as well as outdoor adventure education [17], adventure holidays, and outdoor adventure activities. However, the common denominator is the quest for dynamic, interactive and physically engaging vacations in which tourists play an active role and become the focus of the travel experience as they are not merely content with "watching" but prefer to "participate." This "tourism megatrend" has been identified in several developed countries since the early 1980s [18].

The physical sports activities that comprise OST are usually not conventional or mainstream. Nor are they generally institutionalized, given that they are not subject to codified or standardized sports rules. Moreover, practice is usually self-organized, non-competitive, and often independent. As a result, OST activities are more comparable to recreational games and cannot be considered sports in the traditional sense of the term, but rather as leisure and sports tourism activities [19]. Consequently, they can be regarded as a new tourism modality within

the context of outdoor recreation that includes a wide variety and number of activities, such as outdoor sports and other outdoor activities that involve greater risks and physical challenges for users (rafting, rock climbing, surfing, etc.), what are known as adventure sports or outdoor adventure recreation in the English-speaking world [20]. Outdoor sports or practices in nature [21], new sports leisure activities [22], California sports [23], extreme sports and adventure sports [24], outdoor sports [25], board sports [26], and leisure and sports activities [27], among others, are just some of the many names used to describe these activities. All of these terms convey the immense complexity and variety of OST activities, as well as their different forms, modalities, practices, experiences, and spatial implications depending on the scale and the contexts in which they are practiced. The increasingly blurred lines between the concepts of leisure, tourism, sport, travel, and adventure as a result of this continual transgression of boundaries explains, to a large extent, the internal complexity and diversity of this commercial tourism product.

Moreover, the emergence and recent development of OST is a consequence of the diversification and extensification in contemporary urban societies of self-organized, non-competitive, and individualized sports models linked to tourism and leisure, which are both hedonistic and an alternative to mass sports [8, 27, 28]. However, the growth of OST is also due to many other factors such as increased leisure time; greater purchasing power of those who practice sports in developed countries; loss of interest in traditional, federated, and regulated sports; greater accessibility for wider audiences due to improved transportation links and better mobility; the fact that no special physical conditions are required; the creation of an adventure sports imaginary; the commercialization of sports holidays [29]; attraction to the natural environment as an ideal venue for the practice of sports activities, and the desire for a certain social differentiation through sports tourism. These factors highlight the postmodern nature of these practices within the framework of the unique relationships of otherness and dialectics that sports enthusiasts establish with nature, leisure spaces, and the cultural environment, which generates dissidence and transgression that result in an "own sports culture" or adapted practices [23] beyond conventional sports and tourism disciplines [30].

In summary, OST is a specialized tourism product with enormous potential that is gaining ground as both a practice and a primary motivation for demand by certain segments. It is also a complementary or secondary motivation for other types of tourism such as rural, cultural, or nature tourism and even some types of accommodation, given that it enhances the destination from the more comprehensive approach of "destination product" [31] and could substantially help to correct economic and demographic imbalances in many depressed areas. From the concept of destination in the 1970s, which was characterized by purely geographic stereotypes, tourism has now become a much more complex concept articulated around a series of experiential sequences that visitors can choose from, in which the practice of outdoor sports is viewed as a recreational product with one of the greatest potentials within the tourism supply of many destinations [32].

In order to obtain a comprehensive vision of OST and empirically analyze business supply behavior and strategies in this tourism subsector, it is first necessary to clearly define the variables of interest, in particular:

1. Regional environment and destination competitiveness, including regional tourism resources that provide the appropriate space and conditions for active tourism practices, products, and services [33]. OST is a phenomenon of irrefutable regional and socioeconomic importance, and a structuring dimension of sporting spaces, places, and cultures in contemporary societies. This type of tourism has diverse identities depending on the

experiences, practices, perceptions, and assessments of the users and stakeholders who create outdoor sports spaces tailored to their needs [34].

2. Internal analysis and business competency. An in-depth analysis should be performed to determine the degree to which the companies have adapted their practices to ensure the optimal management and use of the available regional resources, as well as the competency of the sector's SMEs. The internal and external management model of active tourism companies should also be analyzed according to whether the company is concentrated and integrated within some type of associative or business cluster; individualized and non-concentrated when companies operate in isolation, and non-concentrated but integrated when companies join together in an association but do not exercise joint functions in a destination. Similarly, the innovation needs of the stakeholders should be analyzed in at least three main areas (tourism marketing innovation, product innovation, and technology innovation), as well as user profile and level of satisfaction with the product.

3. Strategic and socioeconomic environment and external factors. Synergies between companies, levels of public-private collaboration and involvement in associations, grouping of interests for purposes of promotion and marketing, subcontracting of services with other companies, levels of mobility of supply and relationships between outdoor sporting spaces and local and regional production systems are other factors that should be analyzed [35].

Based on the above considerations, the research background, and the theoretical framework, this empirical study aims to analyze the behavior, motivations, commercial strategies, and business policies regarding the emerging commercial active tourism supply in a specific tourist destination of southern Spain, the province of Cordoba. In particular, we examine rural and nature tourism segments with ample potential for the practice of outdoor physical sports activities but where demand is limited due to the management, promotion, and marketing deficiencies of the companies and establishments involved. To this end, we analyzed the different products that companies currently offer and the diverse sub-modalities of OST that currently exist in the province as a whole, as well as their levels of quality, problems of viability and sustainability, and level of adaptation to demand, among other variables.

## Methods

For the empirical analysis, data on the existing active tourism companies were obtained from the official Andalusia Tourism Registry database (Regional Government of Andalusia) and the National Classification of Economic Activities (CNAE) for the period 2017–2018, in addition to economic data from the Trade Registry and, when needed, corporate and personal income tax returns and other reports from company websites (Table 1). All active, registered companies in that period were taken into account for the survey, resulting in a total of 44 companies. Companies that were active but not registered were not included in the survey sample (an additional 20%, approximately).

**Table 1. Survey data sheet.**

| | Demand surveys |
|---|---|
| *Population* | Companies registered in the Andalusia Tourism Register database (52 companies) |
| *Sample size* | 44 companies |
| *Sample error* | ± 3.9% |
| *Confidence level* | 95%; p = q = 0.5 |
| *Date of the survey* | 1 October 2017 to 15 October 2018 |

The method used was exploratory descriptive analysis in two stages [36, 37]. In the first stage, data were obtained from the registry of active tourism companies in the province of Cordoba. In the second stage, fieldwork was carried out consisting of 12 in-depth interviews with the main stakeholders involved, each of which lasted an hour on average. Respondents were selected using snowball sampling based on their involvement in the decision-making processes used to resolve each of the issues analyzed. A questionnaire with more than 40 items (mostly structured) was also developed to directly survey the companies. The fieldwork/data collection process began on 1 October 2017 and finalized on 15 October 2018, with 94% of the surveys completed directly and in person, and only 6% online.

Data were obtained for the variables considered most relevant for the study of business behavior and business owners' perception of the OST market, in line with the existing literature and case studies [38]. The survey items were structured into three blocks: a) *resources and regional environments used*, *competition in the tourist destination*, *perception of environmental impacts*, *and sustainability of practices*; b) *internal analysis and business competency* (company's legal form and corporate profile, human resources and professional training, business management model differentiating between concentrated–integrated, non-concentrated–individualized, and non-concentrated–integrated) [39], safety measures, taxonomy and practice of activities, actions and perception of quality, perception of demand from the supply side, user motivations and consumption habits, promotion and marketing strategies, and future business prospects (internal analysis), c) *strategic and socioeconomic environment and incidental external factors* (insertion and behavior in collaborative networks, supply clusters, etc.; synergies and connections with other tourism subsectors; perception of public authorities' promotion policies and future perceptions of the external business environment).

The primary purpose of the analysis was to identify the main variables related to company behavior that would enable the companies to i) redirect and develop a management model suitably integrated into the supply chains of the tourist destinations or places where they provide their services, ii) take advantage of these supply chains to offer high added value via quality products, and iii) ensure the sustainability of micro-SMEs. Business size is an important factor for the survival of organizations and an essential aspect for the analysis of profitability and competitiveness, as well as a variable that has a significant influence on the structure of sports organizations. Two research objectives were proposed to examine the characteristics and behavior of these active tourism companies in depth. The first was to classify and describe companies according to their legal form, geographical location, size, and other relevant factors. The second was to study micro-SMEs' economic and business variables and their opinions and perceptions regarding their evolution and prospects: total assets; operating income; number of employees; operation and management models; relationship to other destination products; anticipated evolution and development; and the opinion and perception of problems, threats, and opportunities in the tourism subsector, among others.

## Results

Based on the survey data, it was determined that most of the active tourism companies in the province of Cordoba are of recent creation: 70.5% were registered in the Andalusia Tourism Registry in the period 2011–2015 and 22.7% in the period 2005–2010. Only 4.5% of companies were created before 2005 and only 2.3% after 2015. A total of 69.10% have 3 or fewer employees, 11.9% have 3–4 employees, and only 19% have more than 4. The results also showed a high level of seasonality and temporality, given that a significant number of employees do not work throughout the year and/or full time.

The analysis of the companies' registered capital highlight their micro nature and marked economic fragility. Around 36% have a registered capital of €0–€3,000, 48% have €3,001–€10,000, 5% have €20,001–€30,000, and only 11% have a registered capital of more than €60,000. However, the evolution of some of the companies' net worth since their incorporation until 2016 was generally positive. Over that period, the number of companies with a capital of more than €60,000 increased by 5%. Additionally, 2% of companies had a capital of €50,001–€60,000 and €30,001–€40,000, when there had previously been none with this capital. These companies, albeit not without difficulties, were also able to ride out the 2008–2011 crisis and increase their capitalization. However, a significant 14% have zero or negative assets and are expected to disappear in the near future. The evolution of the companies has been volatile over the last decade, and although the 2008–2011 crisis was followed in 2012–2014 by a significant period of expansion and recovery, in 2015 the market took a downward turn again.

Despite the fact that these companies are micro-SMEs, their sports instructors and managers have a relatively acceptable level of training and qualifications. A total of 48% of the companies have an instructor specialized in outdoor physical sports activities regulated by Decree 190/1996 of August 2 and 32% have an instructor with a qualification in specific sports modalities (Royal Decree 1913/1997 of December 19) or in tourism animation (Decree 246/2001 of November 6), mostly at the post-secondary vocational level. The interviews and surveys verified that this type of company is considered a very accessible and viable form of employment and business outlet due to the low initial investment required and the perceived relative ease of managing and running a physical sports business, for which sports graduates are specifically prepared. However, this perception may be somewhat subjective as the result of professional bias, given that many establishments encounter serious difficulties in terms of their economic and business management, marketing, and promotion, as well as in meeting customer demand; aspects for which sports graduates have not been sufficiently trained. As highlighted in other case studies [40], the results of our analysis show that the majority of sports graduates who already have some experience in company management perceive a disconnection between the vocational training they receive in the current Spanish educational system (LOGSE), the TECO outdoor physical sports activities qualifications they are required to have, and the actual needs of active tourism companies.

As regards the companies' business management, a large percentage stated that they employ a concentrated–integrated management model (68%), which in principle means that they belong to business associations or supply clusters aimed at covering users' needs in a specific area. However, as the data obtained from the qualitative survey items and direct observation show, membership in business associations such as the National Association of Active Tourism Companies (ANECA) or local or regional associations such as the Tourism Initiative Centers (CITs) is very low and the companies are not involved in supply and marketing clusters. A large percentage of companies' are managed in a non-concentrated–integrated manner (27%), operate in isolation and carry out their activities independently, do not belong to associations or supply clusters, and have very little involvement in the regional tourism supply. In addition, although another 5% do belong to business associations, they recognize that the associations rarely contribute to or facilitate the marketing and supply of combined inter-company products or the creation of supply chains for different quality products.

In relation to the type of activities supplied in the province of Cordoba, there is a predominance of "soft" outdoor activities such as hiking (75%), mountain biking (61%), and equestrian tourism (45%). However, very few activities generate significant added value given that they are linked to other more superior products, macro-products, and diverse heritage resources, which not only makes business competitiveness difficult but also hinders competitiveness in the tourist destination itself. The main complementary services offered are photography and

video shooting, as well as the sale of merchandise, which generate very little income, while only 25% of companies offer tailored or à la carte services, and arrange private transport and transfers.

The viability of these companies is further complicated by the strong seasonality of demand, which is highly concentrated in the months of July (66% of the total), August (45%) and June (59%). These summer months, together with May and September, account for almost all the demand, which is practically non-existent in the autumn and winter months. Added to this strong seasonality is the fact that active tourism primarily attracts young people. Specifically, 42% are aged 17–25 and 32% are aged 26–35, in addition to a significant percentage (16%) of school children aged 8–16. As a result, the public who use these products have medium-low purchasing power and are mainly of local (36%) and domestic origin (57%). However, users do have a strong sense of loyalty towards these activities (50% have previously practiced out-door adventure sports, only 27% are first-time users, and 16% frequently repeat the same activities).

Table 2 shows user motivations for practicing physical sports activities. The majority wants to practice sport and physical exercise in an active, fun, or entertaining way (34%), while others pursue sensations of adventure, strong emotions, or controlled risk (27%), closely followed by those who merely seek direct and interactive contact with nature and a sense of freedom (23%). In general, the study of motivations highlights the marked individualistic, hedonistic, masculine, non-competitive, and unregulated character of outdoor sports practices, which has already been observed in other case studies [9, 11, 28].

According to the active tourism entrepreneurs surveyed, these primary motivations tend to remain high and are a key factor in the significant increase of active tourism supply in advanced and postmodern societies in recent decades. One of the most frequently cited moti-vations is the possibility of experiencing a relatively controlled adventure with medium-low risk, which is accessible to large audiences of different socio-demographic profiles (32% of the responses), followed by the desire for direct and interactive contact with nature as an "escape" for a mainly urban clientele (27% of the responses) and, lastly, the desire for an adrenaline rush via strong emotions and sensations in the open air (25%).

It is also interesting to note the company owners and managers' heightened awareness of environmental and nature protection. As they stated, this is an important pull factor for the recreational tourism demand and a guarantee of sustainability, as the practice of outdoor sports is inextricably linked to nature and requires a well-conserved natural environment. Table 3 shows that 48% consider that the majority of activities do not have a significant on the environment, as they are "soft" and "non-invasive", while 27% consider that the impact is important but in relation to specific activities such as motor sports, the practice of which is very restricted in protected natural spaces [41]. Lastly, few companies are unaware of the envi-ronmental impacts and repercussions of their activities or simply did not respond (5%).

**Table 2. Primary user motivations for practicing outdoor physical sports activities from the perspective of supply.**

| | |
|---|---|
| Practicing sport and physical exercise in an active, fun, and entertaining way | 34% |
| Pursuing feelings of adventure, strong emotions, and controlled risk | 27% |
| Direct contact and interactive enjoyment of nature and a sense of freedom | 23% |
| Pleasure, hedonism | 14% |
| Competitiveness and social relations in group practices | 2% |
| Don't know/no answer | 0% |

**Source:** Survey of active tourism companies in Cordoba compiled by the authors.

**Table 3. Perception of company owners on environmental impact of outdoor physical sports activities.**

| | |
|---|---|
| The natural environment is not significantly impacted by the primary activities provided by the company | 48% |
| Specific types of activities have a significant environmental impact (e.g., motor sports and other more invasive activities) | 27% |
| The activities frequently have a significant environmental impact and damage natural areas | 18% |
| The impact rates are not concerning | 2% |
| Don't know/no answer | 5% |

**Source:** Survey of active tourism companies in Cordoba compiled by the authors.

The survey asked the companies about resources and preferences when selling their products, their commercial relationship with users, the activities they provide, the tourism environment, and local agents, in line with other case studies [33]. The results reflect the tendency of owners to promote their business primarily through advertisements and dissemination on the Internet (30% of cases) and to a lesser extent in specialized magazines (18%), posters (14%), tourism associations and tour operators (11%), tourist information offices (9%), and information leaflets (7%). It is also noteworthy that only 5% advertise or market their products through travel agencies, central reservation offices, and conventional tourism intermediaries (Table 4) due to a lack of business interest in and awareness of the tourism subsector, and the low profit margins for intermediaries when negotiating with active tourism companies.

The survey also aimed to determine the owners and managers' perceptions of the short and medium-term evolution of the active tourism subsector (Table 5). In most cases (66%), the results show that the respondents have a positive perception regarding the evolution of this subsector and foresee a significant growth in supply and demand. In 25% of cases, however, the survey responses were nuanced: the future evolution of active tourism is viewed as being very volatile due to the wide range of activities offered and the diverse behaviors and characteristics they entail, demand profiles, and the implementation of restrictions in conservation areas and other rural and natural areas. Only 5% of the respondents think that this tourism subsector will see a more limited growth in the next few years, with a certain tendency to slow down or stagnate, while 2% stated that these activities experienced their "golden age" some time ago. They believe that these practices will no longer be free, independent, and non-regulated or non-federated, but will be successively subjected to a "sportization" process and officially recognized as federated or regulated sports, thereby losing their identity and limiting the companies' opportunities for expanding to wider audiences.

**Table 4. Resources and preferences when marketing and promoting active tourism products.**

| | |
|---|---|
| Ads, information and advertising on the Internet and social networks | 30% |
| Advertisements and publicity in specialized magazines | 18% |
| Conventional posters | 14% |
| Tourism associations | 11% |
| Tourist information offices and other tourism entities | 9% |
| Printed information leaflets/booklets | 7% |
| Ads on radio stations | 5% |
| Travel agencies, reservation centers, and conventional tourism intermediaries | 5% |
| Press ads | 2% |
| TV commercials | 0% |

**Source:** Survey of active tourism companies in Cordoba compiled by the authors.

**Table 5. Perception of active tourism business owners on the development of the subsector in the short and medium term.**

| | |
|---|---|
| Activities clearly on the rise in general | 66% |
| Very volatile trends depending on the type of activity | 25% |
| Limited growth or stagnation in the short and medium term | 5% |
| The free practice of outdoor sports will be officially regulated and lose their original identity | 2% |
| These practices have had their "golden age" and are now in decline | 2% |
| Other perceptions | 0% |

**Source:** Survey of active tourism companies in Cordoba complied by the authors.

When asked about the active tourism products that could have the highest growth and best performance in terms of future supply and demand, the respondents identified a wide variety of foreseeable trends due to their marked heterogeneity. Bearing in mind that this study focuses on an inland rural area with large extensions of hilly and mountainous areas, as well as protected natural spaces, it comes as no surprise that the respondents perceive a greater evolution and growth of physical sports activities on land; particularly hiking, followed by canyoning and mountaineering. In contrast, there is a negative perception with regard to the future growth of winter sports such as alpine skiing, snowmobiling, and mushing, as well as motor sports such as quad biking and off-road vehicles. The former, as a result of the shortening of the winter sports season and the effects of climate change in areas such as Sierra Nevada [42], and the latter because of the increasingly extensive restrictions and prohibitions to their practice in rural and natural areas. The prospect of growth of some water sports, such as rafting, kayaking, and sailing, is also promising, given that the region has large expanses of water (wetlands and reservoirs, the Guadalquivir River, and some sections of whitewater rivers such as in the Genil River). Air sports could also have a bright future with activities such as paragliding and hang gliding, owing to the favorable weather and topographical conditions with many escarpments between the Sierra Morena and the Guadalquivir Valley in Cordoba.

A bivariate analysis of the active tourism companies yielded the following relationships (Table 6):

**Table 6. Results of the bivariate analysis of the variables from the perspective of the stakeholders offering active tourism in the province of Cordoba.**

| Discrete variables | $\chi^2$ | df | $p$-value | Phi | Approximate significance |
|---|---|---|---|---|---|
| Company legal form/Business management model | 9.313 | 1 | .002 | | |
| Company legal form/Company resources | 7.124 | 1 | .008 | | |
| Company legal form/Turnover in the year 2016 | 5.316 | 1 | .021 | -0.356 | 0.021 |
| Company legal form/Reasons for increased activity | 0.302 | 1 | .583 | | |
| Company legal form/Environmental impact of active tourism | 1.566 | 1 | .211 | 0.189 | 0.211 |
| Gender of the company owner/management model | 0.017 | 1 | .895 | 0.020 | 0.895 |
| Gender of the company owner/User level of outdoor sports | 0.49 | 1 | .826 | | |
| Gender of the company owner/Future of outdoor activities | 16.377 | 1 | < .001 | | |
| User level/User trend | 2.573 | 1 | .109 | -0.242 | 0.109 |
| User level/ecological impact of active tourism products | 20.878 | 1 | < .001 | -0.689 | 0.000 |

$\chi^2$ Chi-squared statistic. Related variables for $\alpha = 0.05$, df = degrees of freedom.

Discrete variables with phi value (25% of the cells count less than 5).

**Source:** The authors.

1. A strong relationship between the company's legal form and business management model ($\chi^2$ = 9.313, $p$ = .002).

2. A relationship between the company's legal form and available resources ($\chi^2$ = 7.124, $p$ = .008).

3. Company's legal form and turnover in 2016 ($\chi^2$ = 5.316, $p$ = .021). Given that 25% of the cells count less than 5, the phi coefficient was used to corroborate the result of the chi-square.

4. Relationship between the company's legal form and turnover in 2016.

5. Relationship between the gender of the company owner and the future of outdoor activities ($\chi^2$ = 16.377, $p <$ .001).

6. User level and ecological impact of active tourism products ($\chi^2$ = 20.878, $p <$ .001).

As regards the qualitative results drawn from the in-depth interviews, the majority of companies stated that they are functionally isolated from the rest of the tourism offering in the province and highlighted the lack of synergies for joint promotion and marketing. In this regard, those interviewed felt that they are left to go it alone and are largely ignored by the administration when it comes to receiving public aid and incentives. They also mentioned issues such as the shortage of infrastructure and facilities for the practice of activities in natural areas, which also suffer from problems of accessibility and are subject to heavy restrictions on their use for recreational tourism activities. A significant number of the companies interviewed also agree that there are scarce resources for the promotion and marketing of their products and few channels to avoid excessive dependence on foreign operators and reservation centers. Lastly, most of the companies seem to agree on the need for a legal and administrative framework to govern these active tourism establishments in a manner that is more in line with the reality and the problems they face so as to promote sustainable growth and prevent situations of intrusion and unfair competition in this tourism sub-sector by unauthorized companies and even public or private entities which offer activities that do not comply with the current legal framework.

## Discussion

One of the initial hypotheses verified in this study is the existence of an overwhelming supply of micro-SMEs with a weak and poorly structured productive and business fabric that is still in need of significant innovation in terms of communication technologies, marketing and commercialization, as well as business management. Such changes are essential if they are to offer a more competitive product and turn this economic subsector into a genuine driver of development for the province's rural areas, whose traditional, primary activities have been structurally affected by a major socioeconomic downturn and crisis. The weak and non-significant relationship between customer value marketing and strategic marketing decisions, as well as the managers' limited knowledge of how to implement quality management parameters, are two of the most visible factors negatively affecting the sustainability of these companies.

Our findings coincide with many studies cited in the literature review regarding the characteristics of active tourism companies in inland areas of Spain. Specifically, that the business fabric consists of small companies, 80% of which, if not more, are microenterprises [2, 12, 13, 37, 38, 43, 44]. Similarly, the results show that in terms of size and legal form, micro- and small enterprises predominate and are mostly limited companies or self-employed enterprises. In

general, 27% and 5% of companies are organized around a non-concentrated–individualized and non-concentrated–integrated business management model, respectively. Moreover, a moderate and directly proportional relationship was found between the companies' legal form and business management model. With regard to risk assessment, we calculated the odds ratio to compare the legal form of the enterprise by category (i.e., companies, associations, and joint ventures vs. self-employed individuals) to the company's management model (i.e., concentrated–integrated vs. non-concentrated–individualized business management). We determined that the self-employed are 8.214 times more likely to employ a non-concentrated–individualized or non-concentrated–integrated management model than companies, associations, or joint ventures.

The differences with respect to the legal form of limited companies and public limited companies are not significant, although the latter is barely represented in the study sample (2%), in which limited companies (45%) and self-employed enterprises (39%) predominate. The functional isolation and individualism of this type of company with regard to the creation and management of its product supply and promotion and marketing is clearly evident. Indeed, most owners continue to view other local companies as the competition rather than as an opportunity for combining efforts and collaborating in the creation of macro-products, shared experiences, or joint promotion and marketing to capture wider markets.

We also found that although the tourism subsector of this study area gradually recovered its total assets, operating income, and net worth after the 2008 crisis, the evolution of these parameters has been somewhat volatile and suffered many ups and downs over the past ten years, with a more recent tendency toward deceleration and stagnation. These data should make us reflect on whether the size of these tourism sports businesses—which are small with limited material and human resources and scarce professional training—is sufficient to cope with times of crisis, as well as the challenges of sustainability, competitiveness, and the level of qualifications required to meet the increasingly challenging demand.

Despite the importance of quality, the fragile economic and business models of these entities does not permit investing in quality management. And although different quality management models for sports organizations and OST emphasize the importance of users' opinions and perceptions for conceptualizing and measuring service quality [45, 46], the results suggest that they are not, at least not systematically, taken into account by the majority of micro-SMEs. In this regard, it makes no sense to speak of total quality if the quality perceived by users and their level of satisfaction and loyalty are not considered. Given the limitations of these companies in terms of material and human resources, there seems to be an urgent need to adopt some type of simple tool that could be used by managers to obtain objective feedback on their daily operations. This would provide an initial diagnosis and help them to identify some of their strengths and weaknesses, as well as aspects that could be improved.

The fact that a considerable number of small companies employ a non-concentrated–integrated business management model, and are therefore isolated from the rest of the tourism supply and other stakeholders in the OST subsector, partly explains why the province lacks an inter-company collaborative network to share experiences and/or tangible resources (information, investment in joint promotion and marketing, employment, etc.). A collaborative network would be essential to create and develop combined products that include the physical sports activities of all the companies involved, as well as to implement strategic promotion and marketing campaigns. Furthermore, given that OST is considered a complementary activity for other tourism segments, collaborative networks should be extended to other tourism subsectors that could benefit from active tourism. The ubiquity of the companies surveyed in the province of Cordoba, which happens to host the majority of active tourism practices and

products, may also facilitate this inter-company collaboration network, particularly in protected natural spaces, such as natural parks, where the supply is heavily concentrated.

Despite its relatively recent emergence in rural inland areas [8–10, 47–49], the globalization and highly competitive environment of this tourism sports subsector require companies to be suitably market-oriented and manage their customer portfolio as a key strategic asset to increase business profitability and achieve a sustainable competitive advantage based on strategic marketing decisions. However, it is important to highlight that this study revealed a significant development regarding the dissemination of information via the Internet, which has displaced advertisements in specialized magazines and other more conventional media. Nonetheless, investment in innovation and new marketing, e-commerce, and direct commercial management technologies aimed at the end user remains scarce.

As many of the responses to the open questions showed, the companies face problems related to the specific legislation and public policies to promote active tourism both in Andalusia and nationally as they do not correspond to their day-to-day business reality and needs. Some of the hurdles they must overcome to be able to work in protected natural areas include excessive administrative procedures, grants and subsidy systems, the promotion of training and capacity building, the fight against "hidden" businesses, liability insurance, unfair competition, and administrative licenses. This is particularly prevalent in natural parks, where, according to the company owners, the Spanish Natural Resource Management Plans and Use and Management Master Plans establish numerous restrictions and prohibitions that are not always well-founded or justified. As has been noted in other studies [18, 25, 27, 41], these plans tend to be very generic in nature, as they are not based on prior environmental assessment studies to determine the real impact of physical sporting activities on the natural environment and do not take into account the types of activities and spaces where they are practiced. Moreover, these restrictions are in place in spite of the strong environmental awareness of both the companies and the clients due to the fact that these natural spaces constitute the base for the practice of sports activities and business sustainability. These problems and conflicts can be explained in part by the absence of a collaborative network for the management of sports activities in protected natural spaces with the participation and consensus of business owners, associations, public authorities, and qualified instructors, among other stakeholders. A collaborative network would clearly have positive effects for achieving more sustainable development, as has already been verified in other studies [50].

## Conclusions

In this article, we have presented a comprehensive study on the characteristics and economic-business structure of a tourism sub-sector that has been little studied in Spain due to its recent development in the last two decades, particularly in rural areas and protected natural spaces that are emerging or incipient tourism destinations. In this sense, we believe that our contributions can serve as a reference—both methodologically and in terms of the content and approaches—for further studies on inland rural areas with similar characteristics and problems. More specifically, our findings can be useful for destinations with a weak network of micro-SMEs that are functionally isolated from the rest of the tourism offering and governed by a somewhat intuitive and poorly oriented market behavior due to the inadequate management of their client portfolio and scarce knowledge of business management practices in general; factors that hinder them from taking advantage of key strategic assets to increase business profitability and achieve a competitive and sustainable market advantage.

Moreover, we have obtained a substantial amount of detailed economic information on active tourism companies in the province of Cordoba, Spain. Although several of the

companies were reticent to provide such data, we were able to complete the information through other sources such as trade registers, professional registers of self-employed workers, and other administrative and official sources. Likewise, we believe that we have achieved the fundamental purpose of this study: to identify the main variables driving these companies' behavior; behaviors that need to be redirected if they are to be successful. This information can be of use to companies interested in implementing a management model properly integrated in the productive chain of the tourism destinations where they provide their services, promoting business intelligence measures to improve the competitiveness of this subsector, or developing new active tourism projects.

This study has also underlined the advantages of tourism chains to create high added value through quality products and ensure the sustainability of these micro-SMEs in the sports tourism sector. The business dimension is an important factor for the survival of organizations and essential for the analysis of profitability and competitiveness, as well as an important variable in sports organization structures. Therefore, an attempt has been made to examine in depth the characteristics and behavior of these active tourism companies in the province, which face serious challenges related to business development and sustainability.

Finally, despite the importance of our results, this study also has some limitations. In particular, the analysis could have taken into account the effects before and after the 2008–2011 crisis on company size. Although our analysis included all company sizes, the indirect data does not seem to indicate significant changes in this regard, with the exception of a higher increase in limited rather than self-employed companies in order to avoid greater economic risks. Moreover, some of the companies operating in the outdoor sports market were excluded from the analysis given that these sports are not their main product, as is the case of environmental education centers [51]. Additionally, some rural accommodation establishments and tourist services are not formally registered as active tourism companies. Therefore, a new line of research accounting for these factors could be undertaken in the future, particularly with regard to unfair competition and the underground economy in this tourism sub-sector, which seems to be a significant problem according to the numerous companies surveyed.

## Supporting information

**S1 File. Survey analysis of active tourism supply and demand in the province of Cordoba, Spain.**
(DOCX)

**S1 Data. Survey data SPSS.**
(SAV)

## Acknowledgments

We thank active tourism companies for answering the surveys and three anonymous reviewers for their constructive comments and suggestions on earlier drafts of this paper.

## Author Contributions

**Conceptualization:** Francisco Molina Navarro, Manuel Rivera Mateos.

**Data curation:** Francisco Molina Navarro, Manuel Rivera Mateos.

**Investigation:** Francisco Molina Navarro, María Genoveva Millán Vázquez de la Torre.

**Methodology:** María Genoveva Millán Vázquez de la Torre.

**Resources:** Manuel Rivera Mateos.

**Software:** Francisco Molina Navarro.

**Supervision:** Manuel Rivera Mateos.

**Validation:** María Genoveva Millán Vázquez de la Torre.

**Writing – original draft:** Francisco Molina Navarro.

**Writing – review & editing:** Manuel Rivera Mateos, María Genoveva Millán Vázquez de la Torre.

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
