## [Decision Letter · Decision Letter 0]

27 Oct 2020

PONE-D-20-27330

OUTDOOR SPORTS AND ACTIVE TOURISM COMPANY MANAGEMENT IN CORDOBA (SOUTHERN SPAIN): AN EMPIRICAL STUDY ON THE PERCEPTION AND BEHAVIOR OF SUPPLY

PLOS ONE

Dear Dr. Millán Vázquez de la Torre,

Thank you for submitting your manuscript to PLOS ONE. After careful consideration, we feel that it has merit but does not fully meet PLOS ONE’s publication criteria as it currently stands. Therefore, we invite you to submit a revised version of the manuscript that addresses the points raised during the review process.

We look forward to receiving your revised manuscript.

Kind regards,

Jesús Peña-Vinces

Academic Editor

PLOS ONE

Journal Requirements:

'The funders had no role in study design, data collection and analysis, decision to publish, or preparation of the manuscript.'

*Please include your amended statements within your cover letter; we will change the online submission form on your behalf.*

Reviewers' comments:

Reviewer's Responses to Questions

**Comments to the Author**

1. Is the manuscript technically sound, and do the data support the conclusions?

Reviewer #1: Yes

Reviewer #2: Yes

Reviewer #3: Partly

2. Has the statistical analysis been performed appropriately and rigorously? 

Reviewer #1: Yes

Reviewer #2: Yes

Reviewer #3: Yes

3. Have the authors made all data underlying the findings in their manuscript fully available?

Reviewer #1: Yes

Reviewer #2: Yes

Reviewer #3: Yes

4. Is the manuscript presented in an intelligible fashion and written in standard English?

Reviewer #1: Yes

Reviewer #2: Yes

Reviewer #3: Yes

5. Review Comments to the Author

Reviewer #1: The article is very interesting, on a relevant topic in the area of tourism, very well written and structured, in addition to having data that correctly supports the conclusions reached by the authors.

Reviewer #2: The objectives and relevance of the work are clearly specified in the introduction of the manuscript. The developed theme centered on the outdoor physical sports activities (OSP) is supported by b well-identified background of the study theme, framework of reference and adequate conceptual too. The authors handle the concepts with precision and place them objectively in the Spanish context, highlighting the potential of tourism based on outdoor physical sports activities (OSP).

The descriptive-exploratory method and in-depth interviews allow the work to be empirically developed and the necessary data collected to plan a well-founded discussion.

The results respond to the objectives set and are well supported by the findings of the work and the methods used.

An objective discussion is established on the management models and the problems faced by companies that promote active tourism (outdoor physical sports activities) in Córdoba, Spain.

The final reflections, as conclusions, provide new knowledge to evaluate the problems and challenges in the management and marketing of outdoor physical sports activities.

The bibliography is extensive, up-to-date and pertinent to develop the general reference framework and particularly in the Spanish context as well as the quantitative analyzes.

Reviewer #3: The methodology indicates that a quantitative-qualitative empirical study was carried out together with in-depth interviews applied to key informants.

However, the results and discussion do not show qualitative results (from the in-depth interviews). They should be included in order to later contrast with the quantitative findings, at least with those obtained in the bivariate analysis of the companies studied.

The article lacks the conclusions section, where the limitations of the work can be highlighted, the new information proposed by the text, synthesizing the relevant results, without mentioning everything said in the same way, rather presenting the same, but in a novel and synthetic way.

Present the projection of the article, that is, new research challenges considering the results obtained, propose lines of work that were not covered in the article

6. PLOS authors have the option to publish the peer review history of their article (what does this mean?). If published, this will include your full peer review and any attached files.

Reviewer #1: No

Reviewer #2: No

Reviewer #3: No

---

## [Author Response · Author response to Decision Letter 0]

3 Nov 2020

Dear Editor,

We would like to thank the reviewers for their comments and suggestions. We have made the changes to the manuscript to address the issues they have raised. 

We are also grateful to Plos One for giving us the opportunity to publish our article. 

Kind regards, 

Genoveva Millán

Reviewer 1

Thank you for reviewing our article. We appreciate the comments made.

Reviewer 2

Thank you for reviewing our article. We appreciate the comments made.

Reviewer 3

We appreciate the suggestions made by Reviewer 3, which have helped to improve our manuscript.

Suggestion 1: However, the results and discussion do not show qualitative results (from the in-depth interviews). They should be included in order to later contrast with the quantitative findings, at least with those obtained in the bivariate analysis of the companies studied.

Following the suggestion of Reviewer 3, a new paragraph has been included in the Results section with some of the qualitative results drawn from the in-depth interviews. The new paragraph reads as follows:

“As regards the qualitative results drawn from the in-depth interviews, the majority of companies stated that they are functionally isolated from the rest of the tourism offering in the province and highlighted the lack of synergies for joint promotion and marketing. In this regard, those interviewed felt that they are left to go it alone and are largely ignored by the administration when it comes to receiving public aid and incentives. They also mentioned issues such as the shortage of infrastructure and facilities for the practice of activities in natural areas, which also suffer from problems of accessibility and are subject to heavy restrictions on their use for recreational tourism activities. A significant number of the companies interviewed also agree that there are scarce resources for the promotion and marketing of their products and few channels to avoid excessive dependence on foreign operators and reservation centers. Lastly, most of the companies seem to agree on the need for a legal and administrative framework to govern these active tourism establishments in a manner that is more in line with the reality and the problems they face so as to promote sustainable growth and prevent situations of intrusion and unfair competition in this tourism sub-sector by unauthorized companies and even public or private entities which offer activities that do not comply with the current legal framework.”

Suggestion 2: The article lacks the conclusions section, where the limitations of the work can be highlighted, the new information proposed by the text, synthesizing the relevant results, without mentioning everything said in the same way, rather presenting the same, but in a novel and synthetic way.

Present the projection of the article, that is, new research challenges considering the results obtained, propose lines of work that were not covered in the article

A Conclusions section has been added to address the observations of Reviewer 3 as follows:

“In this article, we have presented a comprehensive study on the characteristics and economic-business structure of a tourism sub-sector that has been little studied in Spain due to its recent development in the last two decades, particularly in rural areas and protected natural spaces that are emerging or incipient tourism destinations. In this sense, we believe that our contributions can serve as a reference—both methodologically and in terms of the content and approaches—for further studies on inland rural areas with similar characteristics and problems. More specifically, our findings can be useful for destinations with a weak network of micro-SMEs that are functionally isolated from the rest of the tourism offering and governed by a somewhat intuitive and poorly oriented market behavior due to the inadequate management of their client portfolio and scarce knowledge of business management practices in general; factors that hinder them from taking advantage of key strategic assets to increase business profitability and achieve a competitive and sustainable market advantage.

Moreover, we have obtained a substantial amount of detailed economic information on active tourism companies in the province of Cordoba, Spain. Although several of the companies were reticent to provide such data, we were able to complete the information through other sources such as trade registers, professional registers of self-employed workers, and other administrative and official sources. Likewise, we believe that we have achieved the fundamental purpose of this study: to identify the main variables driving these companies’ behavior; behaviors that need to be redirected if they are to be successful. This information can be of use to companies interested in implementing a management model properly integrated in the productive chain of the tourism destinations where they provide their services, promoting business intelligence measures to improve the competitiveness of this subsector, or developing new active tourism projects.

This study has also underlined the advantages of tourism chains to create high added value through quality products and ensure the sustainability of these micro-SMEs in the sports tourism sector. The business dimension is an important factor for the survival of organizations and essential for the analysis of profitability and competitiveness, as well as an important variable in sports organization structures. Therefore, an attempt has been made to examine in depth the characteristics and behavior of these active tourism companies in the province, which face serious challenges related to business development and sustainability.”

To address the observation of Reviewer 3 regarding the study limitations, the following paragraph, which was originally at the end of the Discussion section in the first version of the article, has been moved to the Conclusions section.

“Finally, despite the importance of our results, this study also has some limitations. In particular, the analysis could have taken into account the effects before and after the 2008–2011 crisis on company size. Although our analysis included all company sizes, the indirect data does not seem to indicate significant changes in this regard, with the exception of a higher increase in limited rather than self-employed companies in order to avoid greater economic risks. Moreover, some of the companies operating in the outdoor sports market were excluded from the analysis given that these sports are not their main product, as is the case of environmental education centers [51]. Additionally, some rural accommodation establishments and tourist services are not formally registered as active tourism companies. Therefore, a new line of research accounting for these factors could be undertaken in the future, particularly with regard to unfair competition and the underground economy in this tourism sub-sector, which seems to be a significant problem according to the numerous companies surveyed.”

---

## [Decision Letter · Decision Letter 1]

25 Nov 2020

OUTDOOR SPORTS AND ACTIVE TOURISM COMPANY MANAGEMENT IN CORDOBA (SOUTHERN SPAIN): AN EMPIRICAL STUDY ON THE PERCEPTION AND BEHAVIOR OF SUPPLY

PONE-D-20-27330R1

Dear Dr. Millán Vázquez de la Torre,

We’re pleased to inform you that your manuscript has been judged scientifically suitable for publication and will be formally accepted for publication once it meets all outstanding technical requirements.

Kind regards,

Jesús Peña-Vinces

Academic Editor

PLOS ONE

Reviewer's Responses to Questions

**Comments to the Author**

1. If the authors have adequately addressed your comments raised in a previous round of review and you feel that this manuscript is now acceptable for publication, you may indicate that here to bypass the “Comments to the Author” section, enter your conflict of interest statement in the “Confidential to Editor” section, and submit your "Accept" recommendation.

Reviewer #3: All comments have been addressed

2. Is the manuscript technically sound, and do the data support the conclusions?

Reviewer #3: Yes

3. Has the statistical analysis been performed appropriately and rigorously? 

Reviewer #3: Yes

4. Have the authors made all data underlying the findings in their manuscript fully available?

Reviewer #3: Yes

5. Is the manuscript presented in an intelligible fashion and written in standard English?

Reviewer #3: Yes

6. Review Comments to the Author

Reviewer #3: Dear authors:

The observations made in the first review were considered and incorporated into the manuscript. The article demonstrates scientific rigor but is also a relevant topic for the geographical context.

7. PLOS authors have the option to publish the peer review history of their article (what does this mean?). If published, this will include your full peer review and any attached files.

Reviewer #3: No

---

## [Editor Report · Acceptance letter]

1 Dec 2020

PONE-D-20-27330R1 

Outdoor sports and active tourism company management in Cordoba (southern Spain): an empirical study on the perception and behavior of supply 

Dear Dr. Millán Vázquez de la Torre:

I'm pleased to inform you that your manuscript has been deemed suitable for publication in PLOS ONE. Congratulations! Your manuscript is now with our production department. 

Kind regards, 

on behalf of

Dr. Jesús Peña-Vinces 

Academic Editor

PLOS ONE